# Leaf Color Chart (LCC)-Based Precision Nitrogen Management for Assessing Phenology, Agrometeorological Indices and Sustainable Yield of Hybrid Maize Genotypes under Temperate Climate

Suhail Fayaz [1], Raihana Habib Kanth [2], Tauseef Ahmad Bhat [2,*], Mohammad Valipour [3,*], Rashid Iqbal [4], Awais Munir [5], Aijaz Nazir [6], Mohd Salim Mir [2], Shafat Ahmad Ahanger [7], Ibrahim Al-Ashkar [8] and Ayman El Sabagh [9]

[1] Department of Agronomy, School of Agriculture, Lovely Professional University, Ludhiana 144411, PB, India
[2] Division of Agronomy, Faculty of Agriculture, Sher-e-Kashmir University of Agricultural Sciences and Technology of Kashmir, Wadura, Sopore 193201, Kashmir, India
[3] Department of Engineering and Engineering Technology, Metropolitan State University of Denver, Denver, CO 80217, USA
[4] Department of Agronomy, Faculty of Agriculture and Environment, The Islamia University of Bahawalpur, Punjab 63100, Pakistan
[5] Institute of Agro-Industry and Environment, The Islamia University of Bahawalpur, Punjab 63100, Pakistan
[6] Agromet Unit, Faculty of Horticulture, Sher-e-Kashmir University of Agricultural Sciences and Technology of Kashmir, Shalimar, Srinagar 190025, Kashmir, India
[7] Division of Plant Pathology, Faculty of Agriculture, Sher-e-Kashmir University of Agricultural Sciences and Technology of Kashmir, Wadura, Sopore 193201, Kashmir, India
[8] Department of Plant Production, College of Food and Agriculture, King Saud University, Riyadh 11451, Saudi Arabia
[9] Department of Agronomy, Faculty of Agriculture, Kafrelsheikh University, Kafrelsheikh 33156, Egypt
* Correspondence: tauseekk@gmail.com (T.A.B.); mvalipou@msudenver.edu (M.V.)

**Abstract:** Excessive nitrogenous fertilization in years resulted in larger nitrogen and profit losses. This problem can be reduced by using need-based and time-specific nitrogen management. Therefore, a field experiment was carried out during the Kharif season of 2019 and 2020 in order to evaluate the impact of precision nitrogen management on the phenology, yield and agrometeorological indices of hybrid maize genotypes at the Agronomy Research Farm, FoA Wadura, Sopore, SKUAST-Kashmir. The experiment was carried out in split-plot design consisting of maize hybrids (Shalimar Maize Hybrid-2 Vivek-45 and Kanchan-517) as main plot treatments and precision nitrogen management ($T_1$: Control, $T_2$: Recommended N, $T_3$: 25% N as basal $\leq$ LCC 3@20 kg N ha$^{-1}$, $T_4$: 25% N as basal $\leq$ LCC 3@30 kg N ha$^{-1}$, $T_5$: 25% N as basal $\leq$ LCC 4@20 kg N ha$^{-1}$, $T_6$: 25% N as basal $\leq$ LCC 4@30 kg N ha$^{-1}$, $T_7$: 25% N as basal $\leq$ LCC 5@20 kg N ha$^{-1}$ and $T_8$: 25% N as basal $\leq$ LCC 5@30 kg N ha$^{-1}$) as sub-plot treatments. Results demonstrated that maize hybrids showed a non-significant difference in attaining different phenophases during both years. However, Shalimar Maize Hybrid-2 demonstrated higher grain (62.35 and 60.65 q ha$^{-1}$) and biological yield (170.26 and 165.86 q ha$^{-1}$), a higher number of days to attain different phenological stages in comparison to Vivek-45 and Kanchan-517 thereby achieved higher heat units, PTUs, HTUs, PTI. The application of nitrogen through LCC $\leq$ 5@30 kg N ha$^{-1}$ noted higher grain yield (61.27 and 59.13 q ha$^{-1}$) and biological yield (171.30 and 166.13 q ha$^{-1}$) during 2019 and 2020 respectively. Higher values of Growing degree days (GDD), Heliothermal units (HTU), Photothermal units (PTU), Phenothermal index (PTI), heat use efficiency (HUE) and radiation use efficiency (RUE) were observed in the application of nitrogen through LCC $\leq$ 5@30 kg N ha$^{-1}$ and required the highest number of days to reach different phenophases than other treatments during crop growing seasons of 2019 and 2020. The results demonstrated that Nitrogen application based on LCC $\leq$ 5@30 proved effective and should be adopted in maize hybrids especially in Shalimar Maize Hybrid-2 to attain higher yield under the temperate climate of Kashmir Valley.

**Keywords:** agrometeorological indices; heat use efficiency; N application; phenophases; sustainable yield

## 1. Introduction

Maize (*Zea mays* L.), the most vibrant food grain crop is grown under distinct soil and climatic conditions. Temperature is considered one of the detrimental factors for maize production. Temperature-based agro-meteorological indices viz. growing degree days (GDD), heliothermal units (HTU), photothermal units (PTU), phenothermal index (PTI) and heat use efficiency (HUE) show a linear relationship between phenophasic development with base and optimum temperature [1]. Being a thermophilic plant, it is highly sensitive to lower temperatures at all developmental stages [2]. An optimal average temperature of 20–22 °C is required for the entire growing season of maize crop. Maize is a very demanding crop due to its high need for nutrients from the soil, particularly nitrogen, phosphorus and potassium. Nitrogen is a vital part of amino acids, the principal components of proteins, as well as a part of the DNA molecule; therefore it is crucial for cell division and reproduction [3]. Nitrogen is the principal ingredient for maximizing maize production. However, excessive nitrogen application can harm the ecosystem and has the potential to leak into subterranean water. It is critical to manage nitrogen to fit the crop's needs [4]. For attaining self-sufficiency in food grain production and safeguarding food security, efficient and effective nutrient management has to play a pivotal role in achieving this goal. Precise and responsive nitrogen fertilizer management in respect of maize crops is crucial from both economic as well as environmental points of view. Excessive fertilization in years resulting from static and blind fertilizer recommendations leads to huge nutrient and profit losses especially nutrient nitrogen. This problem can be reduced by offering smart and precise time-specific advice [5]. Fertilizer management at the proper dose and at the right time is critical to the efficient use of fertilizers [6]. According to a new analysis of yield patterns in numerous long-term studies, maize yields are either stagnant or decreasing. There are also reports of large yield gaps between research trials and farmer's fields, which, if correctly adjusted, may increase present maize production enormously. This has mostly been linked to poor agro-management techniques, with incorrect nutrient management playing a key role. A new set of nutrient management concepts like Site-specific nutrient management (SSNM) is aiming to meet the nutrient requirements of crops grown in the field in a smart, accurate and precise way. It aims for use of indigenous sources of nutrients like crop wastes, residues and manures, applying nutrient fertilizers at optimum and precise rates and at critical growth stages to tighten the gap between a high-yielding crop's nutrient requirements and the indigenous nutrient supply [7]. Tools like Leaf Color Chart (LCC) have considerably aided in predicting the timing and rate of nutrient demand and numerous studies have indicated increased nutrient usage efficiency and yield enhancement. A leaf color chart is a collection of color swatches that may be compared to a leaf in similar lighting circumstances [8]. The International Rice Research Institute and the Philippine Rice Research Institute in collaboration created a leaf color chart that guides farmers to regulate nitrogen application in rice fields based on crop demand. The technique is low-cost, making it accessible for use to even the most resource-constrained farmers [9]. LCC being a simple and non-destructive technique provides quick and reliable monitoring of leaf greenness by the visual appearance of spectral characteristics of leaves and might be an efficient guide to farmers for efficient and time-specific application of N. To assess efficient and smart Nitrogen management in a range of conditions, including soil, climate, variety, management and so on, particularly in rice, the LCC is widely used [10]. Blanket recommendation of fertilizer application is widely used without taking into consideration the spatial and temporal soil variability and crop demand leading to very low and large yield gaps.

The developmental change and biomass accumulation in crops are majorly determined by the climatic variables during the crop growing period, which primarily affects the heat unit requirement of the crop from one developmental phase to another [11,12], however, no significant research was conducted to find the phenological changes due to varied heat unit requirement in different maize hybrids with the application of nitrogen through LCC for efficient utilization of weather parameters. Therefore, the study was carried out to analyze the yield and phasic development of hybrid maize genotypes in response to the nitrogen application through LCC by assessing the agro-meteorological indices and thermal efficiencies.

## 2. Materials and Methods

### 2.1. Site Description

A field experiment was carried out at Crop Research Farm of Agronomy, Faculty of Agriculture, Wadura Sopore, SKUAST-Kashmir during Kharif 2019 and 2020, located between 34°21′ N latitude and 74°23′ E longitude having an altitude of 1590 masl. The experimental area had unvaried topography with good drainage facilities. The experimental trial (Table 1) was silty clay loam in texture and medium in organic carbon (0.66%) and the available nitrogen (320.5 kg ha$^{-1}$), phosphorus (19.75 kg ha$^{-1}$) and potassium (170.20 kg ha$^{-1}$) were medium with neutral pH (7.1).

**Table 1.** Physico-chemical parameters of the soil of experimental trial.

| | Characteristics | Status | Range | Method Used |
|---|---|---|---|---|
| A. | Physical Texture | | | International Pipette Method [13] |
| | Coarse sand | 20.00 | | |
| | Silt (%) | 50.00 | | |
| | Clay (%) | 30.00 | | |
| | Texture | | Silty-clay–loam | |
| B. | Chemical Analysis | | | |
| | PH | 7.1 | Neutral | Blackman's glass method [14] |
| | OC | 0.66 (%) | Medium | Black and Walkely method [15] |
| | N | 320.5 (kg ha$^{-1}$) | Medium | Potassium permanganate method [16] |
| | P | 19.75 (kg ha$^{-1}$) | Medium | Extraction with 0.5 M NaoHCO$_3$ [17] |
| | K | 170.2 (kg ha$^{-1}$) | Medium | Flame photometer method [14] |

### 2.2. Weather Conditions

The weather was found variable throughout the entire growing period of the crop. The maximum temperature varied from 23.36 to 31.93 °C and 23.14 to 33.94 °C and minimum temperature from 9.57 to 18.29 °C and 6.71 to 18.96 °C and the average maximum relative humidity ranged from 64.43 to 91.00 percent and 59.71 to 87.71 percent, whereas mean minimum relative humidity varied from 38.57 to 70.43 percent and 36.57 to 75.29 percent during the crop growing period of 2019 and 2020, respectively (Figures 1 and 2). The crop received 371.5 mm and 303.0 mm of precipitation during the crop growth seasons of 2019 and 2020, respectively.

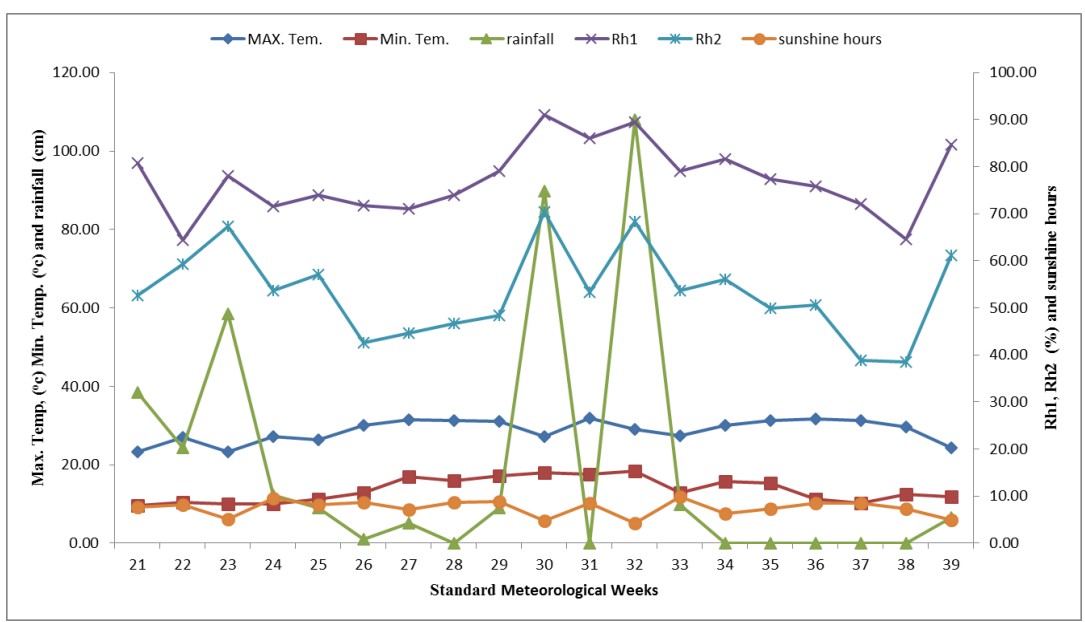

**Figure 1.** Mean meteorological data during maize crop growth season of 2019. Where, SSH means Sunshine hours, and $RH_1$ and $RH_2$ means maximum and minimum relative humidity.

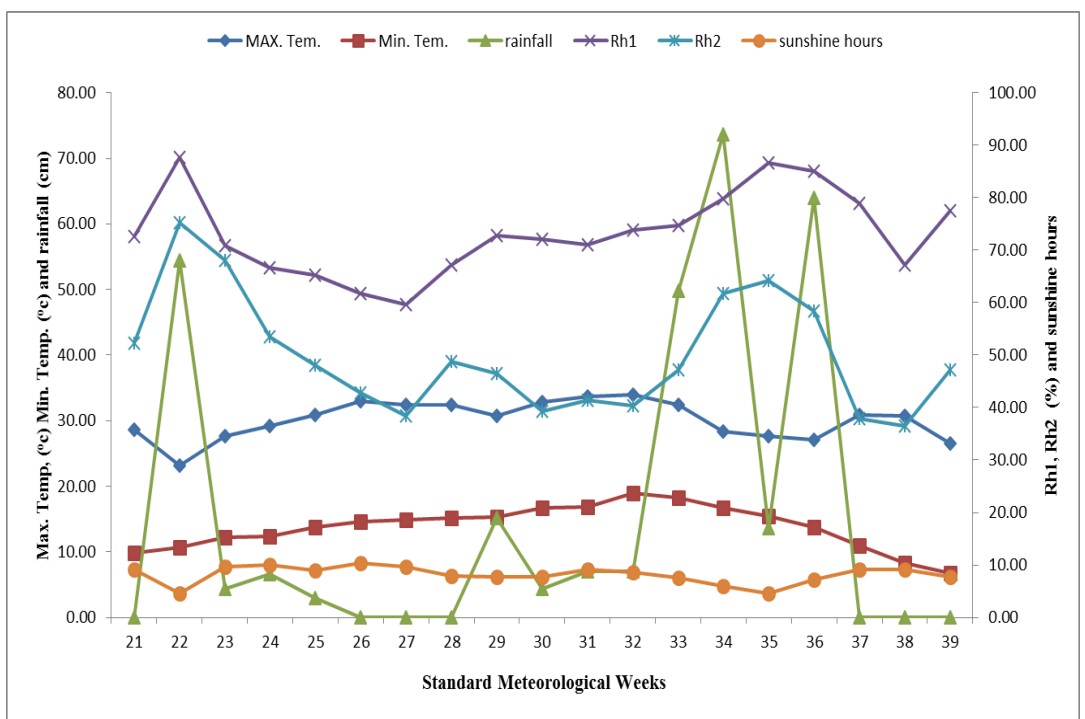

**Figure 2.** Mean meteorological data during maize crop growth season of 2020. Where, SSH means Sunshine hours, $RH_1$ and $RH_2$ means maximum and minimum relative humidity.

*2.3. Experimental Design and Treatment Details*

The experiment consisted of two factors with three maize hybrids as main plot treatments viz. (Shalimar Maize Hybrid-2, Vivek-45 and Kanchan-517) and eight rates of nitrogen application viz. ($T_1$: Control, $T_2$: Recommended N, $T_3$: 25% N as basal $\leq$ LCC 3@20 kg N ha$^{-1}$, $T_4$: 25% N as basal $\leq$ LCC 3@30 kg N ha$^{-1}$, $T_5$: 25% N as basal $\leq$ LCC 4@20 kg N ha$^{-1}$, $T_6$: 25% N as basal $\leq$ LCC 4@30 kg N ha$^{-1}$, $T_7$: 25% N as basal $\leq$ LCC 5@20 kg N ha$^{-1}$ and $T_8$: 25% N as basal $\leq$ LCC 5@30 kg N ha$^{-1}$) as sub-sub-plot treat-

ments, replicated thrice and set up in split plot design. Vivek-45 maize hybrid was released by VPKS Almora, India whereas the Kanchan-517 is a private maize hybrid released by Kanchanganga Seed Company in Hyderabad. Further, the Shalimar Maize Hybrid-2 is a local hybrid released by SKUAST-Kashmir for the temperate climate of Kashmir.

### 2.4. Crop Management Practices

As per the main plot and sub-plot treatments, the experimental field was prepared. Each main plot of maize hybrids was divided into eight sub-plots to accommodate eight nitrogen management practices. Two ploughings were given with a tractor to obtain desirable depth and were followed by leveling. Replication borders, main-plot and sub-plot bunds were made manually. At sowing time, phosphorous and potassium were applied as basal dose @60 and 40 kg $P_2O_5$ and $K_2O$ ha$^{-1}$ respectively. By employing Leaf Color Chart (LCC), nutrient nitrogen was supplied as per the treatments and in Recommended N treatment, that is, 150 kg ha$^{-1}$, half of the nitrogen was applied as basal dose, and the rest of the half was applied at the knee-high and tasseling stages of the crop in two equal splits. The nutrients, nitrogen, phosphorous and potassium were supplied as Urea, DAP (Diammonium Phosphate) and MOP (Muriate of Potash), respectively. Before the seed sowing operation, it was ensured that sufficient moisture for the germination of the seed is present in the soil. Presoaked and treated seed with Bavistin + Captan 1:1 ratio@2 kg$^{-1}$ seed for Turcicum leaf blight and Metalaxyl 35 SD@4 g Kg$^{-1}$ for brown strip downy mildew was sown in the lines with a spacing of 75 cm × 20 cm. After sowing the lines were closed with soil and slightly pressed so as to have good contact of the seed with the soil. The irrigation was applied in the field, as and when needed by the crop. Pre-emergence herbicide application in the form of atrazine (0.75 kg a.i. ha$^{-1}$) at 3 DAS followed by manual weeding at 21 DAS was carried out for complete removal of weeds. A solution of chlorpyriphos (0.05%) was sprayed once on the standing crops as a prophylactic measure to control insect pests. Net plot area was taken after removing the two border rows for eliminating the border effect in the experimental plots. For yield assessment, maize crop samples were collected from the net plot area and the harvesting was carried out manually by hand and hand shelling was carried out for the separation of grains from cobs.

### 2.5. LCC-Based Nitrogen Application

Leaf Color Chart, being easy to handle, is a favorable and non-destructive tool for the determination of the need-based nitrogen demand of rice. LCC comprised six color strips (from yellowish green to dark green) fabricated with veins matching the leaves of maize crops. LCC was developed by International Rice Research Institute (IRRI) in collaboration with Philippine Rice Research Institute (Manila). LCC was employed in the experimental field for real-time nitrogen management with five green strips ranging from yellow-green to dark green with each shade designated by 1, 2, 3, 4 and 5. Starting from 21 days after sowing till starting of the tasseling stage, LCC readings were taken at every four-day interval. In each plot from the sampling area, 10 healthy plants were taken at random. From each plant, the top-most completely expanded leaf was selected and the LCC readings were noted by keeping the middle part of the leaf on the chart, and the leaf color was determined by blocking the sunlight with the body to avoid errors in leaf color reading (Figure S1). Nitrogen was applied as per the treatment whenever the green color of more than 5 out of 10 leaves was determined to be equal to or below a set critical limit of LCC score. The final split application of Nitrogen was carried out by 78 DAS coinciding with the tasseling stage (Table 2).

**Table 2.** Treatment wise splits and quantity of nitrogen applied in maize crop during 2019 and 2020.

| Treatments | No. of Splits | 2019 | | | 2020 | | |
|---|---|---|---|---|---|---|---|
| | | Shalimar Maize Hybrid-2 | Vivek-45 | Kanchan-517 | Shalimar Maize Hybrid-2 | Vivek-45 | Kanchan-517 |
| Control | - | - | - | - | - | - | - |
| Recommended N | 3 | 150 | 150 | 150 | 150 | 150 | 150 |
| LCC $\leq$ 3@20 kg N ha$^{-1}$ | 4 | 80 | 80 | 80 | 80 | 80 | 80 |
| LCC $\leq$ 3@30 kg N ha$^{-1}$ | 3 | 90 | 90 | 90 | 90 | 90 | 90 |
| LCC $\leq$ 4@20 kg N ha$^{-1}$ | 5 | 100 | 100 | 100 | 100 | 100 | 100 |
| LCC $\leq$ 4@30 kg N ha$^{-1}$ | 4 | 120 | 120 | 120 | 120 | 120 | 120 |
| LCC $\leq$ 5@20 kg N ha$^{-1}$ | 6 | 120 | 120 | 120 | 120 | 120 | 120 |
| LCC $\leq$ 5@30 kg N ha$^{-1}$ | 5 | 150 | 150 | 150 | 150 | 150 | 150 |

### 2.6. Biometric Crop Observations

The readings for the number of days to reach different phenophases were noted when the plants crossed 50 percent of that particular phenophase. Grain yield was recorded after separating cobs from stalk and husk. From each net plot area, all the cobs were sun-dried and the grains were separated by hand shelling. The grain yield was regulated to 15 percent moisture content and was taken in kg ha$^{-1}$ and then expressed as q ha$^{-1}$ (quantil per hectare). For recording biological yield, the weight of the bundle taken from each net plot after harvesting was noted after sun drying for 3–4 days and was accordingly converted into q ha$^{-1}$.

### 2.7. Computation of Agrometeorological Indices and Thermal Use Efficiencies

The weather parameters viz. daily max. and min. temperature, sunshine hours, day length and average relative humidity was assessed and employed for the calculation of agrometeorological indices viz. growing degree days (GDD), heliothermal units (HTU), photothermal units (PTU) and phenothermal index (PTI) at different phenophases of the crop using the formulas given below. GDD, HTU, PTU and PTI were measured from the date of sowing to each phenological stage with a base temperature of 10 °C [18].

$$GDD = [(T_{Max} + T_{Min})/2] - T_b \tag{1}$$

where:

$$T_{Max} = \text{Daily max. Temperature}$$

$$T_{Min} = \text{Daily min. Temperature}$$

$T_b$ = base temperature (a temperature below which no development occurs for a given plant species).

$$HTU = (GDD \times SSH) \tag{2}$$

where, SSH (hour) is the daily sunshine hours

$$PTU = (GDD \times DL) \tag{3}$$

DL (hour) is the day length

$$PTI = (\text{Heat units consumed between two phenophases})/(\text{Duration between two phenophases}) \tag{4}$$

$$HUE = (\text{Seed yield (kg/ha)})/(\text{Accumulated heat units (°C)}) \tag{5}$$

$$RUE = (\text{Seed yield (kg/ha)})/(\text{Accumulated heat units (°C)}) \tag{6}$$

For obtaining HUE and RUE, grain yield was divided by the respective accumulated heat units.

*2.8. Statistical Analysis*

Crop phenology and grain yield data were analyzed as per the analysis of variance using the statistical procedure of the split-plot design. The extent of the relationship between yield (Grain/biological) with different indices was computed by means of regression. Regression of yield with respective indices was worked out in the form of a regression equation to find the response of yield (Grain/biological) determined by various indices. The statistical procedure was done employing SPSS software, version 27.0.

**3. Results**

*3.1. Phenology*

Data from the experiment revealed significant variation in maize hybrids with varying rates of N application for the completion of different growth stages (Table 3). The developmental rate of plant growth quantitatively depends on the prevailing temperature. Shalimar Maize Hybrid-2 took the highest number of days (128 and 126) when compared to Vivek-45 and Kanchan-517 for completing different phenophases.

Nitrogen application in maize hybrids through LCC $\leq$ 5@30 and 20 kg N ha$^{-1}$ required the highest number of days to reach the knee-high stage, tasseling stage, silking stage and harvesting in comparison to other treatments during both years of experimentation. However, LCC $\leq$ 4@20 kg N ha$^{-1}$ treatment was found statistically at par with the application of N through LCC $\leq$ 3@30 and 20 kg N ha$^{-1}$ and recommended Nitrogen treatment required a higher number of days to reach different phenophases as compared to control during 2019 and 2020. Furthermore, LCC $\leq$ 5@30 and 20 kg N ha$^{-1}$ consumed the highest value of heat units (GDD) in comparison to other LCC treatments and recommended nitrogen treatment at all phenological stages. Control treatment recorded the lowest number of days to complete different phenophases.

**Table 3.** Effect of precision nitrogen management through LCC on days taken to different phenophases of different hybrid maize genotypes.

| Treatments | Knee-High Stage | | Tasseling Stage | | Silking Stage | | Harvest | |
|---|---|---|---|---|---|---|---|---|
| | 2019 | 2020 | 2019 | 2020 | 2019 | 2020 | 2019 | 2020 |
| **Hybrids** | | | | | | | | |
| Shalimar Maize Hybrid-2 | 41 | 39 | 74 | 73 | 80 | 78 | 128 | 126 |
| Vivek-45 | 39 | 37 | 73 | 71 | 79 | 77 | 127 | 125 |
| Kanchan-517 | 37 | 36 | 72 | 71 | 78 | 76 | 125 | 124 |
| SEm± | 0.49 | 0.39 | 0.42 | 0.40 | 0.65 | 0.61 | 0.90 | 0.96 |
| C.D. (5%) | NS | NS | NS | NS | NS | NS | NS | NS |
| **Nitrogen management** | | | | | | | | |
| Control | 36 | 35 | 69 | 68 | 74 | 72 | 121 | 120 |
| Recommended N | 38 | 37 | 73 | 71 | 79 | 77 | 126 | 124 |
| LCC $\leq$ 3@20 kg N ha$^{-1}$ | 37 | 36 | 70 | 69 | 77 | 75 | 125 | 123 |
| LCC $\leq$ 3@30 kg N ha$^{-1}$ | 38 | 36 | 71 | 70 | 77 | 75 | 125 | 124 |
| LCC $\leq$ 4@20 kg N ha$^{-1}$ | 39 | 38 | 74 | 72 | 79 | 77 | 127 | 125 |
| LCC $\leq$ 4@30 kg N ha$^{-1}$ | 40 | 38 | 74 | 73 | 80 | 79 | 129 | 127 |
| LCC $\leq$ 5@20 kg N ha$^{-1}$ | 40 | 39 | 74 | 73 | 81 | 79 | 129 | 127 |
| LCC $\leq$ 5@30 kg N ha$^{-1}$ | 43 | 42 | 78 | 76 | 85 | 83 | 132 | 130 |
| SEm± | 0.40 | 0.36 | 0.80 | 0.76 | 0.86 | 0.80 | 0.83 | 0.78 |
| C.D. (5%) | 1.22 | 1.09 | 2.41 | 2.29 | 2.58 | 2.40 | 2.49 | 2.34 |
| Interaction | NS | NS | NS | NS | NS | NS | NS | NS |

## 3.2. Agrometeorological Indices

### 3.2.1. Growing Degree Days

The growing degree day concept assesses the relationship between growth and temperature by accounting for the heat unit requirement for the completion of different development stages of the crop. Varying rates of N application showed significant variation in heat unit accumulation among the maize hybrids at different phenological stages during both years (Table 4). Shalimar Maize Hybrid-2 consumed more heat units or GDD (2721.95 and 2790.15 °C) in comparison to Vivek-45 and Kanchan-517 to reach different phenological stages. Nitrogen application of nitrogen through LCC $\leq$ 5@30 kg N ha$^{-1}$ consumed the highest number of heat units (2791.2 and 2869.9 °C), whereas control treatment consumed the lowest heat units. LCC $\leq$ 5@20 kg N ha$^{-1}$ and LCC $\leq$ 4@30 kg N ha$^{-1}$ treatments were statistically at par in consuming heat units but significantly superior to LCC $\leq$ 4@20 kg N ha$^{-1}$, LCC $\leq$ 3@30 and 20 kg N ha$^{-1}$ and recommended N treatment during both the years of experimentation.

**Table 4.** Effect of precision nitrogen management through LCC on GDD (°C) of different hybrid maize genotypes.

| Treatments | Knee-High Stage | | Tasseling Stage | | Silking Stage | | Harvest | |
|---|---|---|---|---|---|---|---|---|
| | 2019 | 2020 | 2019 | 2020 | 2019 | 2020 | 2019 | 2020 |
| **Hybrids** | | | | | | | | |
| Shalimar Maize Hybrid-2 | 735.25 | 773.15 | 1515.75 | 1587.3 | 1684 | 1715.35 | 2721.95 | 2790.15 |
| Vivek-45 | 695.4 | 729.4 | 1492 | 1538.55 | 1629.5 | 1688.05 | 2690.7 | 2771.15 |
| Kanchan-517 | 652.3 | 705.65 | 1468.3 | 1538.55 | 1601.5 | 1662.3 | 2645.95 | 2751.15 |
| SEm$\pm$ | 9.26 | 7.83 | 7.32 | 8.88 | 6.66 | 8.29 | 9.91 | 5.99 |
| C.D. (5%) | 27.80 | 23.50 | 21.97 | 26.65 | 19.98 | 24.88 | 29.72 | 17.88 |
| **Nitrogen management** | | | | | | | | |
| Control | 614.8 | 659.9 | 1372.05 | 1437.05 | 1492 | 1538.45 | 2543.45 | 2653.55 |
| Recommended N | 680.55 | 729.4 | 1492 | 1539.65 | 1642 | 1688.05 | 2668.45 | 2763.74 |
| LCC $\leq$ 3@20 kg N ha$^{-1}$ | 652.3 | 690.04 | 1423.3 | 1488.8 | 1592 | 1634.51 | 2645.9 | 2731.4 |
| LCC $\leq$ 3@30 kg N ha$^{-1}$ | 673.15 | 705.55 | 1445.3 | 1513.3 | 1610.9 | 1639.3 | 2666.95 | 2751.15 |
| LCC $\leq$ 4@20 kg N ha$^{-1}$ | 695.83 | 734.1 | 1512.75 | 1569.55 | 1645.5 | 1688.05 | 2690.7 | 2771.15 |
| LCC $\leq$ 4@30 kg N ha$^{-1}$ | 715.3 | 750.4 | 1515.95 | 1589.3 | 1664 | 1730.1 | 2733.45 | 2809.15 |
| LCC $\leq$ 5@20 kg N ha$^{-1}$ | 739.3 | 773.1 | 1556.3 | 1626.85 | 1695.6 | 1743.1 | 2749.51 | 2816.42 |
| LCC $\leq$ 5@30 kg N ha$^{-1}$ | 783.3 | 846.1 | 1618.5 | 1673.92 | 1784 | 1847 | 2791.2 | 2869.9 |
| SEm$\pm$ | 12.12 | 13.97 | 16.74 | 15.11 | 22.11 | 23.43 | 18.82 | 17.18 |
| C.D. (5%) | 36.47 | 41.92 | 50.24 | 45.34 | 66.34 | 71.29 | 56.46 | 51.54 |
| Interaction | NS | NS | NS | NS | NS | NS | NS | NS |

### 3.2.2. Heliothermal Units (HTU)

Differential rates of N application of maize hybrids recorded significant differences in heliothermal units at various phenophases (Table 5). The three maize hybrids differed significantly with respect to heliothermal units during both crop growth seasons of 2019 and 2020. The highest numbers of heliothermal units (20,614.52 and 22,877.02 °C day hour) were noted under Shalimar Maize Hybrid-2 in comparison to Vivek-45 and Kanchan-517 at different phenological stages.

Data demonstrated that nitrogen application through LCC $\leq$ 5@30 kg N ha$^{-1}$ achieved the maximum number of heliothermal units (21,212.18 and 23,705.37 °C day hour), whereas control treatment recorded the lowest number of heliothermal units, during 2019 and 2020, respectively. The number of heliothermal units observed in LCC $\leq$ 5@20 kg N ha$^{-1}$ and LCC $\leq$ 4@30 kg N ha$^{-1}$ treatments were statistically at par but significantly superior to LCC $\leq$ 4@20 kg N ha$^{-1}$, LCC $\leq$ 3@30 and 20 kg N ha$^{-1}$ and recommended N treatment during crop growth season of 2019 and 2020.

**Table 5.** Effect of precision nitrogen management through LCC on Heliothermal (°C day hour) units of different hybrid maize genotypes.

| Treatments | Knee-High Stage | | Tasseling Stage | | Silking Stage | | Harvest | |
|---|---|---|---|---|---|---|---|---|
| | **2019** | **2020** | **2019** | **2020** | **2019** | **2020** | **2019** | **2020** |
| **Hybrids** | | | | | | | | |
| Shalimar Maize Hybrid-2 | 5737.64 | 6666.93 | 11,785.98 | 13,707.75 | 12,697.36 | 14,811.7 | 20,614.52 | 22,877.02 |
| Vivek-45 | 5342.10 | 6268.90 | 11,461.83 | 13,236.3 | 12,441.95 | 14,561.12 | 20,474.74 | 22,707.91 |
| Kanchan-517 | 5059.38 | 5999.99 | 11,216.18 | 13,211.99 | 12,400.95 | 14,312.35 | 20,206.59 | 22,524.21 |
| SEm± | 77.29 | 63.18 | 90.76 | 95.16 | 76.07 | 73.32 | 46.33 | 51.87 |
| C.D. (5%) | 231.87 | 189.56 | 272.28 | 285.50 | 228.21 | 219.98 | 138.99 | 155.63 |
| **Nitrogen management** | | | | | | | | |
| Control | 4737.38 | 5616.69 | 10,479.28 | 12,449.5 | 11,437.19 | 13,170.82 | 19,328.12 | 21,622.01 |
| Recommended N | 5365.60 | 6228.69 | 11,429.13 | 13,234.66 | 12,537.4 | 14,532 | 20,377.64 | 22,666.85 |
| LCC ≤ 3@20 kg N ha$^{-1}$ | 5098.52 | 5867.26 | 10,871.98 | 12,851.15 | 12,285.28 | 14,032.81 | 20,206.21 | 22,316.39 |
| LCC ≤ 3@30 kg N ha$^{-1}$ | 5307.26 | 5999.13 | 11,071.81 | 13,051.13 | 12,431.13 | 14,073.94 | 20,366.96 | 22,563.49 |
| LCC ≤ 4@20 kg N ha$^{-1}$ | 5340.53 | 6268.83 | 11,716.75 | 13,439.27 | 12,564.12 | 14,532 | 20,475.04 | 22,679.65 |
| LCC ≤ 4@30 kg N ha$^{-1}$ | 5425.94 | 6408.02 | 11,741.54 | 13,661.45 | 12,546.56 | 14,887.62 | 20,675.76 | 23,001.68 |
| LCC ≤ 5@20 kg N ha$^{-1}$ | 5603.60 | 6666.50 | 12,054.06 | 13,984.22 | 12,731.65 | 15,154.62 | 20,813.75 | 23,061.21 |
| LCC ≤ 5@30 kg N ha$^{-1}$ | 6159.43 | 7429.16 | 12,539.44 | 14,411.13 | 13,746.95 | 16,110.01 | 21,212.18 | 23,705.37 |
| SEm± | 108.63 | 81.45 | 132.22 | 125.66 | 166.08 | 174.40 | 132.22 | 138.58 |
| C.D. (5%) | 325.90 | 244.36 | 396.67 | 376.98 | 498.26 | 523.22 | 396.67 | 415.76 |
| Interaction | NS | NS | NS | NS | NS | NS | NS | NS |

### 3.2.3. Photothermal Units (PTU)

The results demonstrated significant variation at different growth stages in photothermal unit accumulation among hybrids under varied N applications (Table 6). The highest number of photothermal units (39,601.65 and 40,549 °C day hour) were observed under Shalimar Maize Hybrid-2 in comparison to Vivek-45 and Kanchan-517, during the crop growing season of 2019 and 2020, respectively. Application of nitrogen through LCC ≤ 5@30 kg N ha$^{-1}$ recorded the highest number of photothermal units (40,581.63 and 41,688.17 °C day hour) during both years. The number of photothermal units recorded by LCC ≤ 5@20 kg N ha$^{-1}$ and LCC ≤ 4@30 kg N ha$^{-1}$ treatments were found at par but significantly superior to LCC ≤ 4@20 kg N ha$^{-1}$, LCC ≤ 3@30 and 20 kg N ha$^{-1}$ and recommended N treatment during both the years, whereas control treatment recorded the lowest number of photothermal units.

**Table 6.** Effect of precision nitrogen management through LCC on Photothermal (°C day hour) units of different hybrid maize genotypes.

| Treatments | Knee-High Stage | | Tasseling Stage | | Silking Stage | | Harvest | |
|---|---|---|---|---|---|---|---|---|
| | **2019** | **2020** | **2019** | **2020** | **2019** | **2020** | **2019** | **2020** |
| **Hybrids** | | | | | | | | |
| Shalimar Maize Hybrid-2 | 10,185.42 | 10,698.08 | 21,397.84 | 22,458.71 | 23,853.86 | 24,201.87 | 39,601.65 | 40,549.25 |
| Vivek-45 | 9622.25 | 10,081.04 | 21,050.63 | 21,682.79 | 23,068.83 | 23,870.72 | 39,125.47 | 40,250.95 |
| Kanchan-517 | 9015.43 | 9747.143 | 20,704.5 | 21,682.79 | 22,659.62 | 23,493.29 | 38,432.42 | 39,938.44 |
| SEm± | 110.04 | 116.18 | 104.32 | 140.81 | 132.48 | 108.93 | 135.08 | 95.38 |
| C.D. (5%) | 330.13 | 348.56 | 312.98 | 422.43 | 397.45 | 326.81 | 405.24 | 286.14 |
| **Nitrogen management** | | | | | | | | |
| Control | 8506.99 | 91,22.238 | 19,358.25 | 20,279.17 | 21,122.24 | 21,734.71 | 36,977.52 | 38,542.81 |
| Recommended N | 9416.77 | 10,082.98 | 21,050.63 | 21,727.03 | 23,245.79 | 23,848.21 | 38,794.82 | 40,143.32 |
| LCC ≤ 3@20 kg N ha$^{-1}$ | 9025.88 | 9538.883 | 20,081.34 | 21,009.45 | 22,537.94 | 23,091.81 | 38,466.98 | 39,673.59 |
| LCC ≤ 3@30 kg N ha$^{-1}$ | 9314.38 | 9753.288 | 20,391.74 | 21,355.19 | 22,805.51 | 23,159.48 | 38,773.01 | 39,960.45 |
| LCC ≤ 4@20 kg N ha$^{-1}$ | 9628.20 | 10,147.95 | 21,343.39 | 22,148.97 | 23,295.34 | 23,848.21 | 39,118.29 | 40,250.95 |
| LCC ≤ 4@30 kg N ha$^{-1}$ | 9897.61 | 10,373.28 | 21,388.54 | 22,427.67 | 23,557.25 | 24,442.28 | 39,739.81 | 40,802.9 |
| LCC ≤ 5@20 kg N ha$^{-1}$ | 10,229.69 | 10,687.08 | 21,957.84 | 22,957.56 | 24,004.61 | 24,625.94 | 39,973.29 | 40,908.5 |
| LCC ≤ 5@30 kg N ha$^{-1}$ | 10,838.52 | 11,696.2 | 22,835.42 | 23,625.15 | 25,284.4 | 26,093.8 | 40,581.63 | 41,688.17 |
| SEm± | 178.42 | 134.77 | 172.32 | 200.76 | 217.40 | 199.72 | 196.25 | 183.30 |
| C.D. (5%) | 535.27 | 404.32 | 516.98 | 602.28 | 652.21 | 599.18 | 588.76 | 549.92 |
| Interaction | NS | NS | NS | NS | NS | NS | NS | NS |

### 3.2.4. Phenothermal Index (PTI)

Degree days per growth day generally expressed as phenothermal index showed an increasing trend from knee high to flowering stage and thereafter showed decreasing trend till maturation suggesting a decline in daily thermal consumption towards senescence. The phenothermal index for consecutive phonological stages of maize hybrids varied significantly under different rates of N application (Table 7). The highest phenothermal index was observed in Shalimar Maize Hybrid-2 at different phenological stages in comparison to Vivek-45 and Kanchan-517 hybrids. However, the application of nitrogen through LCC $\leq$ 5@30 kg N ha$^{-1}$ showed the highest phenothermal index, whereas the lowest phenothermal index was found under control treatment at all phenophases. The phenothermal index values recorded by LCC $\leq$ 5@20 kg N ha$^{-1}$ were significantly superior to LCC $\leq$ 4@30 kg N ha$^{-1}$, LCC $\leq$ 4@20 kg N ha$^{-1}$, LCC $\leq$ 3@30 and 20 kg N ha$^{-1}$ treatments and recommended N treatment during 2019 and 2020, respectively.

**Table 7.** Effect of precision nitrogen management through LCC on Phenothermal index ($^{\circ}$C day day$^{-1}$) of different hybrid maize genotypes.

| Treatments | Knee-High Stage | | Tasseling Stage | | Silking Stage | | Harvest | |
|---|---|---|---|---|---|---|---|---|
| | **2019** | **2020** | **2019** | **2020** | **2019** | **2020** | **2019** | **2020** |
| **Hybrids** | | | | | | | | |
| Shalimar Maize Hybrid-2 | 18.16 | 19.67 | 20.46 | 21.86 | 20.80 | 21.89 | 21.27 | 22.07 |
| Vivek-45 | 17.97 | 19.46 | 20.55 | 21.67 | 20.88 | 21.96 | 21.23 | 22.15 |
| Kanchan-517 | 17.40 | 19.53 | 20.34 | 21.81 | 20.72 | 21.77 | 21.10 | 22.24 |
| SEm$\pm$ | 0.05 | 0.06 | 0.04 | 0.03 | 0.01 | 0.02 | 0.01 | 0.02 |
| C.D. (5%) | 0.15 | 0.18 | 0.11 | 0.09 | 0.04 | 0.06 | 0.03 | 0.07 |
| **Nitrogen management** | | | | | | | | |
| Control | 17.03 | 18.94 | 19.85 | 21.28 | 20.17 | 21.28 | 20.99 | 22.19 |
| Recommended N | 17.62 | 19.75 | 20.44 | 21.55 | 20.90 | 21.97 | 21.19 | 22.14 |
| LCC $\leq$ 3@20 kg N ha$^{-1}$ | 17.44 | 19.53 | 20.22 | 21.63 | 20.79 | 21.86 | 21.18 | 22.15 |
| LCC $\leq$ 3@30 kg N ha$^{-1}$ | 17.86 | 19.38 | 20.24 | 21.65 | 20.70 | 21.76 | 21.13 | 22.26 |
| LCC $\leq$ 4@20 kg N ha$^{-1}$ | 17.93 | 20.04 | 20.54 | 21.64 | 20.74 | 21.80 | 21.19 | 22.11 |
| LCC $\leq$ 4@30 kg N ha$^{-1}$ | 18.06 | 19.62 | 20.46 | 21.89 | 20.72 | 22.17 | 21.27 | 22.13 |
| LCC $\leq$ 5@20 kg N ha$^{-1}$ | 17.74 | 19.80 | 20.44 | 21.87 | 20.91 | 22.04 | 21.18 | 22.04 |
| LCC $\leq$ 5@30 kg N ha$^{-1}$ | 18.17 | 20.22 | 20.80 | 21.81 | 21.01 | 22.19 | 21.16 | 21.95 |
| SEm$\pm$ | 0.10 | 0.11 | 0.09 | 0.07 | 0.03 | 0.05 | 0.05 | 0.04 |
| C.D. (5%) | 0.29 | 0.32 | 0.26 | 0.22 | 0.10 | 0.15 | 0.14 | 0.12 |
| Interaction | NS | NS | NS | NS | NS | NS | NS | NS |

### 3.2.5. Heat Use Efficiency (HUE) and Radiation Use Efficiency (RUE)

The data on heat use efficiency on the basis of grain and biological yield presented in Table 8 demonstrates that the three maize hybrids differed significantly with respect to heat use efficiency and radiation use efficiency on grain and biological yield basis during both years. The highest heat use efficiency of 2.29 and 2.17 kg/ha/$^{\circ}$C day on a grain yield basis and 6.26 and 5.94 kg/ha/$^{\circ}$C day on a biological yield basis was recorded under Shalimar Maize Hybrid-2 in comparison to Vivek-45 and Kanchan-517 at different phenophases. Further, the highest radiation use efficiency of 3.45 and 3.56 kg/ha/$^{\circ}$C day was recorded under Shalimar Maize Hybrid-2 in comparison to Vivek-45 and Kanchan-517 hybrids. Data demonstrated that nitrogen application through LCC $\leq$ 5@30 kg N ha$^{-1}$ observed the highest heat use efficiency of 2.20 and 2.06 (kg/ha/$^{\circ}$C day) on a grain yield basis and 6.14 and 5.79 (kg/ha/$^{\circ}$C day) on the basis of biological yield, whereas lowest heat use efficiency on grain and biological yield basis was recorded under control treatment during 2019 and 2020, respectively. The heat use efficiency on grain and biological yield basis recorded in LCC $\leq$ 5@20 kg N ha$^{-1}$ and LCC $\leq$ 4@30 kg N ha$^{-1}$ treatments were found statistically at par but significantly superior to LCC $\leq$ 4@20 kg N ha$^{-1}$, LCC $\leq$ 3@30

and 20 kg N ha$^{-1}$ and recommended N treatment during crop growth season of 2019 and 2020. The data demonstrated that nitrogen management through recommended N treatment recorded the highest radiation use efficiency of 3.55 and 3.71 kg/ha/°C day, during 2019 and 2020, respectively. The radiation use efficiency recorded by applying nitrogen through LCC ≤ 5@30 and 20 kg N ha$^{-1}$ and LCC ≤ 4@30 kg N ha$^{-1}$ treatments were found superior to LCC ≤ 4@20 kg N ha$^{-1}$ and LCC ≤ 3@30 and 20 kg N ha$^{-1}$ during 2019 and 2020. LCC ≤ 3@20 kg N ha$^{-1}$ recorded higher HUE and RUE than other LCC scores and the lowest HUE and RUE were obtained with LCC ≤ 5@30 and 20 kg N ha$^{-1}$. However, the lowest radiation use efficiency of 3.22 and 3.34 kg/ha/°C day was recorded under control treatment.

**Table 8.** Effect of precision nitrogen management through LCC on heat use efficiency on grain and biological yield basis and radiation use efficiency of different hybrid maize genotypes.

| Treatments | Heat Use Efficiency Grain Yield Basis (kg/ha/°C day) | | Heat Use Efficiency Biological Yield Basis (kg/ha/°C day) | | Radiation Use Efficiency (kg/ha/°C day) | |
|---|---|---|---|---|---|---|
| | **2019** | **2020** | **2019** | **2020** | **2019** | **2020** |
| **Hybrids** | | | | | | |
| Shalimar Maize Hybrid-2 | 2.29 | 2.17 | 6.26 | 5.94 | 3.45 | 3.56 |
| Vivek-45 | 1.93 | 1.79 | 5.59 | 5.24 | 3.40 | 3.52 |
| Kanchan-517 | 1.66 | 1.60 | 5.39 | 4.94 | 3.28 | 3.39 |
| SEm± | 0.03 | 0.04 | 0.05 | 0.16 | 0.01 | 0.01 |
| C.D. (5%) | 0.10 | 0.11 | 0.15 | 0.17 | 0.03 | 0.04 |
| **Nitrogen management** | | | | | | |
| Control | 1.60 | 1.45 | 4.87 | 4.48 | 3.22 | 3.34 |
| Recommended N | 1.95 | 1.81 | 5.74 | 5.36 | 3.55 | 3.71 |
| LCC ≤ 3@20 kg N ha$^{-1}$ | 1.95 | 1.81 | 5.62 | 5.26 | 3.41 | 3.56 |
| LCC ≤ 3@30 kg N ha$^{-1}$ | 1.97 | 1.83 | 5.68 | 5.32 | 3.38 | 3.52 |
| LCC ≤ 4@20 kg N ha$^{-1}$ | 2.07 | 1.93 | 5.92 | 5.56 | 3.41 | 3.50 |
| LCC ≤ 4@30 kg N ha$^{-1}$ | 2.06 | 1.93 | 5.88 | 5.54 | 3.36 | 3.45 |
| LCC ≤ 5@20 kg N ha$^{-1}$ | 2.12 | 2.00 | 5.97 | 5.64 | 3.33 | 3.43 |
| LCC ≤ 5@30 kg N ha$^{-1}$ | 2.20 | 2.06 | 6.14 | 5.79 | 3.34 | 3.44 |
| SEm± | 0.04 | 0.05 | 0.07 | 0.08 | 0.04 | 0.03 |
| C.D. (5%) | 0.13 | 0.14 | 0.22 | 0.23 | 0.09 | 0.08 |
| Interaction | NS | NS | NS | NS | NS | NS |

### 3.3. Yield

Data shown in Table 9 demonstrated that the yield of maize hybrids was governed significantly by variable rates of N application. Shalimar Maize Hybrid-2 achieved the highest grain yield of 62.35 and 60.65 q ha$^{-1}$ as compared to Vivek-45 with grain yield of 51.92 and 49.69 q ha$^{-1}$ and Kanchan-517 with grain yield of 46.42 and 43.92 q ha$^{-1}$ during 2019 and 2020, respectively. Grain yield revealed a significant difference at variable LCC scores. It was noticed that nitrogen management through LCC ≤ 5@30 kg N ha$^{-1}$ recorded maximum grain yield of 61.27 and 59.13 q ha$^{-1}$ during 2019 and 2020, respectively. Similarly, LCC ≤ 5@20 and LCC ≤ 4@30 and 20 kg N ha$^{-1}$ treatments recorded significantly higher grain yield in comparison to treatments with LCC ≤ 3@30 and 20 kg N ha$^{-1}$ and recommended N treatment during the crop growth seasons of 2019 and 2020. Furthermore, nitrogen application through LCC ≤ 3@30 and 20 kg N ha$^{-1}$ was found to be statistically at par with recommended N treatment but superior to control treatment during both years of experimentation.

Data revealed that significant variation was observed in three hybrids with respect to biological yield. Shalimar Maize Hybrid-2 recorded the highest biological yield of 170.26 and 165.86 q ha$^{-1}$ as compared to Vivek-45 (150.47 and 145.16 q ha$^{-1}$) and Kanchan-517 (141.61 and 135.81 q ha$^{-1}$), during 2019 and 2020, respectively. Shalimar Maize Hybrid-2 also revealed significantly higher dry matter than Vivek-45 and Kanchan-517 contributing

to higher biological yield. Nitrogen management through LCC significantly impacted the biological yield during the crop growing season of 2019 and 2020. Data revealed that the highest biological yield of 171.30 and 166.13 q ha$^{-1}$ was achieved with LCC $\leq$ 5@30 kg N ha$^{-1}$, being superior over LCC $\leq$ 5@20 kg N ha$^{-1}$, LCC $\leq$ 4@30 and 20 kg N ha$^{-1}$, LCC $\leq$ 3@30 and 20 kg N ha$^{-1}$ and recommended N treatment during 2019 and 2020, respectively. However, the lowest biological yield of 123.94 and 118.76 q ha$^{-1}$ was noted under control treatment during 2019 and 2020 respectively.

**Table 9.** Effect of real-time nitrogen management through LCC on yield of different hybrid maize genotypes.

| Treatments | Grain Yield (q ha$^{-1}$) | | Biological Yield (q ha$^{-1}$) | |
|---|---|---|---|---|
| | 2019 | 2020 | 2019 | 2020 |
| **Hybrids** | | | | |
| Shalimar Maize Hybrid-2 | 62.35 | 60.65 | 170.26 | 165.86 |
| Vivek-45 | 51.92 | 49.69 | 150.47 | 145.16 |
| Kanchan-517 | 46.42 | 43.92 | 141.61 | 135.81 |
| SEm± | 1.16 | 1.29 | 3.18 | 3.13 |
| C.D. (5%) | 4.45 | 3.85 | 9.60 | 9.49 |
| **Nitrogen management** | | | | |
| Control | 40.65 | 37.50 | 123.94 | 117.76 |
| Recommended N | 52.10 | 49.95 | 153.25 | 148.08 |
| LCC $\leq$ 3@20 kg N ha$^{-1}$ | 51.70 | 49.56 | 148.75 | 143.58 |
| LCC $\leq$ 3@30 kg N ha$^{-1}$ | 52.46 | 50.32 | 151.58 | 146.41 |
| LCC $\leq$ 4@20 kg N ha$^{-1}$ | 55.62 | 53.48 | 159.22 | 154.04 |
| LCC $\leq$ 4@30 kg N ha$^{-1}$ | 56.32 | 54.17 | 160.75 | 155.58 |
| LCC $\leq$ 5@20 kg N ha$^{-1}$ | 58.38 | 56.24 | 164.13 | 158.96 |
| LCC $\leq$ 5@30 kg N ha$^{-1}$ | 61.27 | 59.13 | 171.30 | 166.13 |
| SEm± | 0.91 | 0.78 | 1.02 | 1.02 |
| C.D. (5%) | 2.62 | 2.34 | 2.93 | 2.86 |
| Interaction | S | S | NS | NS |

*3.4. Regression Analysis*

According to the regression coefficient grain yield was influenced by agrometeorological indices and thermal use efficiencies (Figures 3 and 4). The coefficient of determination for grain yield and biological yield ranged from 0.950 to 0.953 and 0.951 to 0.954 with agrometeorological indices respectively. The coefficient of determination was highly significant for grain yield and biological yield with GDD (0.953 and 0.952), HTU (0.950 and 0.954) and PTU (0.953 and 0.951). The deviations in GDD could be attributed to the degree of 95%, respectively. Further, agrometeorological indices viz. HTU and PTU, both recorded variability of 95% with grain yield as well as with biological yield. The coefficient of determination for grain yield with heat use efficiency and radiation use efficiency was 0.99 and 0.05. The variations in HUE and RUE could be explained to the extent of 99% and 5%, respectively. Moreover, the coefficient of determination for biological yield with heat use efficiency (0.99) and radiation use efficiency (0.07) was also found significant. Variations in heat use efficiency and radiation use efficiency could be explained to the extent of 99% and 7%, respectively. Heat use efficiency accounted for maximum variability of 99% for both grain and biological yield.

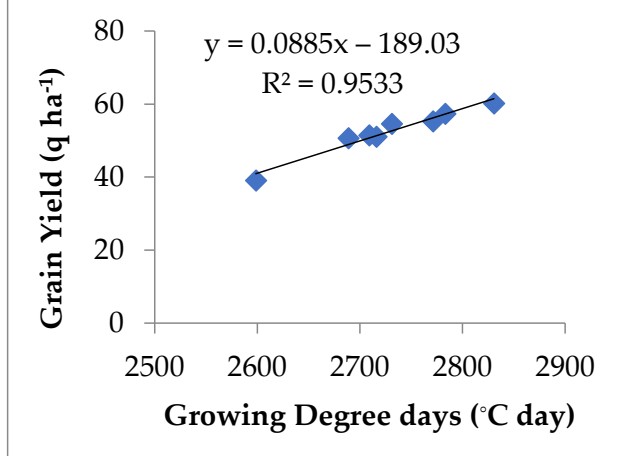

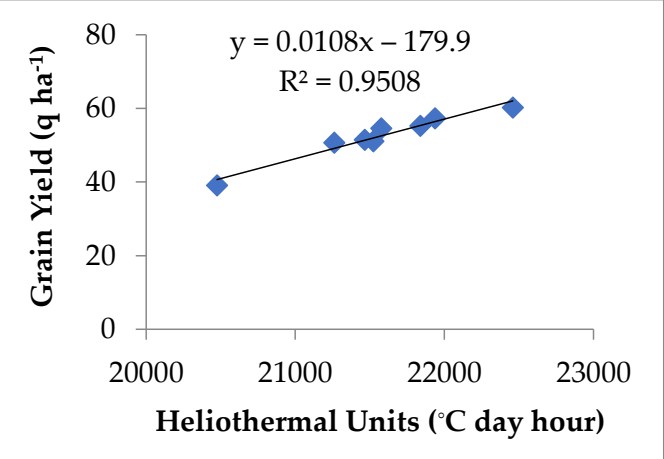

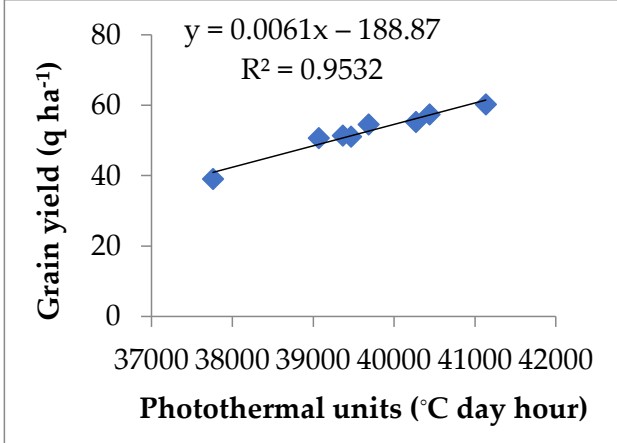

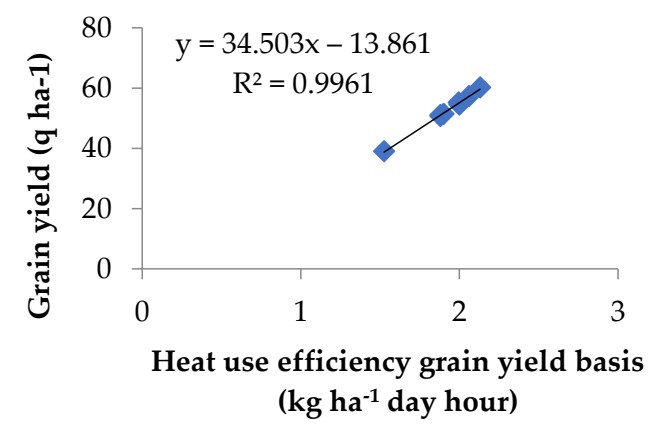

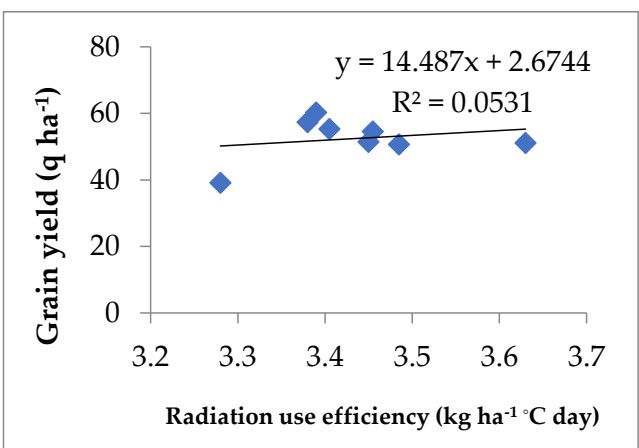

**Figure 3.** Linear regression equation to find the effect of agrometeorological indices on grain yield of hybrid maize.

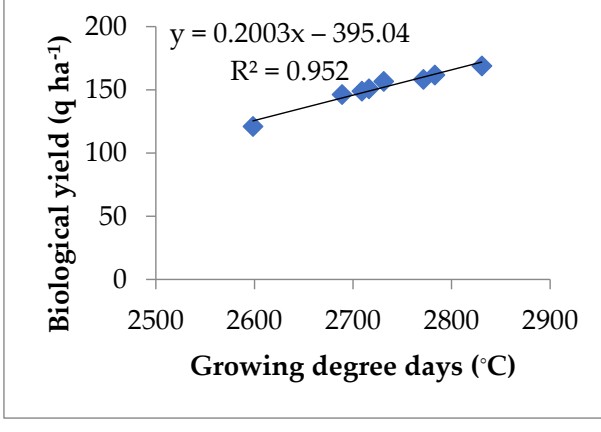
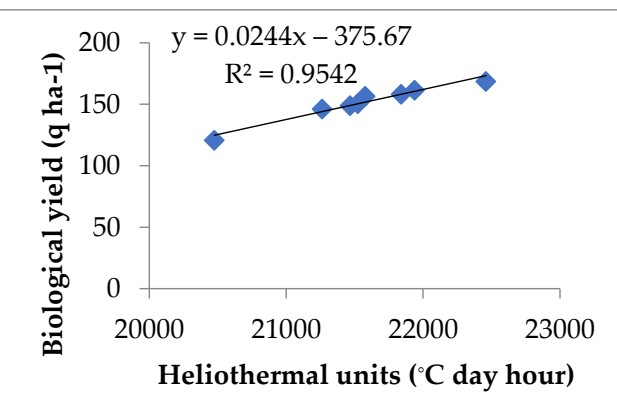
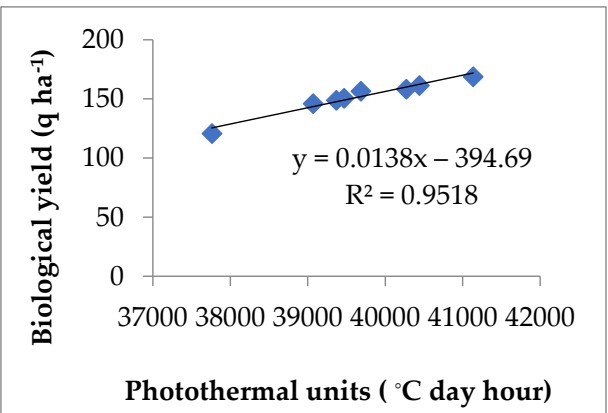
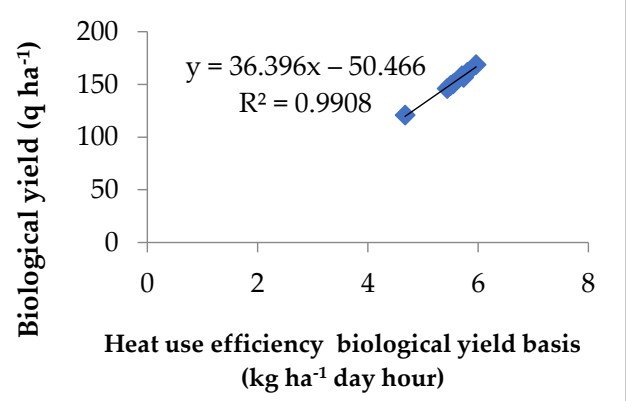
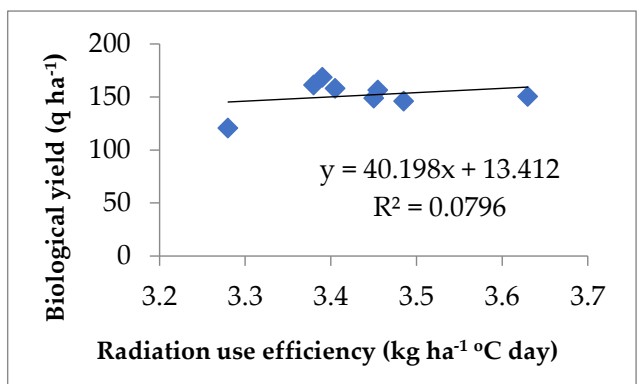

**Figure 4.** Linear regression equation to find the effect of agrometeorological indices on biological yield of hybrid maize.

## 4. Discussion

The differential behavior of maize hybrids to heat unit requirement (GDD), HTU, PTU and PTI may be attributed to their genetic constitution. Furthermore, it may be due to the extended crop growth period of Shalimar Maize Hybrid-2 in the field in comparison to Vivek-45 and Kanchana-517 leading to higher GDD, HTU, PTU and PTI. Also, variation in values of heat use efficiency and radiation use efficiency is attributed to the higher yield of Shalimar Maize Hybrid-2 in comparison to Vivek-45 and Kanchan-517. Our results are in close conformity with the research work of Majumder et al. (2016) [19]; Deshmukh et al. (2021) [20]; Thavaprakash et al. (2007) [21] and Mittal (1996) [22]. A higher number of days required to reach different growth stages in LCC $\leq$ 5@30 and 20 kg N ha$^{-1}$ treatment may be attributed to the higher application of nitrogen that had increased the crop growth cycle and thermal time which in turn increased the GDD of the

crop [23–26]. Furthermore, higher values of HUE and RUE in LCC 5 treatments might be due to the synchronized and increased nitrogen application in a greater number of splits in LCC 5 which accounted for higher heat use efficiency and radiation use efficiency.

For performing need-based Fertilizer applications in cereal crops, especially rice, LCC is a cheap and practically efficient diagnostic tool that has grown significantly in importance for boosting crop productivity and nitrogen recovery. The results revealed that the Shalimar Maize Hybrid-2 variety required a maximum of days to attain different phenological stages and performed better in comparison to Vivek-45 and Kanchan-517, which might possibly be due to variations in their genetic composition. Similar inferences were made by Bhat et al. (2015) [27] while assessing the impact of LCC-based N management on rice varieties. Statistically, insignificant variation was observed in maize hybrids viz. Shalimar Maize Hybrid-2, Vivek-45 and Kanchan-517 in attaining various phenophases, Shalimar Maize Hybrid-2 took more days to reach different phenophases. Shalimar Maize Hybrid-2 consumed more heat units (GDD) to attain different growth phases in comparison to other hybrids, during both years of experimentation. The relationship between crop growth duration and temperature is direct and linear and was explained on the basis of heat unit requirement or the Growing Degree Days concept [18]. The difference in heat unit requirement and days taken to attain different phenophases could be attributed to the genetic makeup of hybrids [28]. Varietal variations for GDD to complete various phenophases have also been observed by Naik et al. (2019) [29], Majumder et al. (2016) [19]; Rajesh et al. (2015) [30]. Due to a timely and balanced supply of nitrogen, there were significant differences between the various N treatments using LCC scores. These variations led to delayed vegetative growth characteristics. When nitrogen was applied based on LCC $\leq$ 5@30 kg N ha$^{-1}$, the crop reached maturity within a greater number of days. In the control case, crop plants transitioned over to the reproductive phase much earlier than LCC score-based treatments, due to the unavailability of nitrogen and thus took the minimum number of days to attain maturity [31–33].

The observed variable yield among the three maize hybrids may be due to their inherent genetic variability. Jyothsna et al. (2021) [34]; Moharana et al. (2017) [35]; Bhat et al. (2015) [36]; Bhavana et al. (2020) [37] and Fayaz et al. (2021) [38] also noticed significant variation while assessing LCC based nitrogen management in different crop varieties. Nitrogen management through LCC $\leq$ 5@30 and 20 kg N ha$^{-1}$ treatments revealed significantly higher grain yield due to an increased amount of nitrogen application with more splits in comparison to other treatments [34]. Also, it fulfilled the demand for crops at different phenophases, thus reducing denitrification and volatilization losses as nitrogen was applied in more splits, adding more grain yield. The improved availability of nitrogen at critical stages increased photosynthate accumulation towards grain coupled with efficient and improved yield contributing characters. The findings are in accordance with Mathukia et al. (2014) [39] and Bhat et al. (2022) [40]. Higher biological yield with nitrogen application through LCC 5 over other LCC scores and recommended N treatment may be due to their favorable impact on the vegetative growth phase [34,36,41].

Nitrogen supply had a significant effect on yield under LCC score-based N management, which could be due to nitrogen impact on the growth in terms of growth characteristics and yield contributing characters, which was in contrast to the control, which had reduced growth characteristics and poor development of yield parameters, resulting in lower yield. In comparison to other treatments, the yield was higher under treatment LCC 5@30 kg N ha$^{-1}$ because timely accessibility of nitrogen increased leaf area, which directly improved dry matter production and subsequently improved growth in yield metrics. Increased grain yield with higher LCC scores highlights the significance of nitrogen in maximizing yield; as a result, the soil's insufficient natural N supply was unable to meet the crop's expanding demand. An increase in the application of N has been linked to a large yield improvement as reported by other researchers also [42,43]. Additionally, the nitrogen need for rice crops changes as the crop grows (from the vegetative to the mature phase), with the highest requirement seen during times of rapid growth. Thus,

applying nitrogen during critical growth stages promotes improved crop development and increases crop yield [44]. The results are in close confirmation with the findings of Ahmad et al. (2016) [45], who observed that application of nutrients at critical stages while considering the inherent capability of soil is important for profitable crop production. The regression analysis demonstrated that GDD accounted for maximum variability which was the foremost agrometeorological index expressing a large effect on grain yield and marking a variation of 95% on both grain and biological yield. Furthermore, heat use efficiency contributed maximum impact on grain and biological yield and contributed a variation of 99% while radiation use efficiency accounted for a minimum contribution of 5% and 7% towards grain yield and biological yield. Fariba et al. (2009) [46]; Pazhanisamy et al. (2020) [47] and Bisma et al. (2020) [48] also found similar results.

## 5. Conclusions

Based on the above results, application of nitrogen through LCC 5 in Shalimar Maize Hybrid-2 required a higher number of days to attain various phonological stages and recorded higher values of accumulated heat units, HTU, PTU, PTI, heat use efficiency and radiation use efficiency and ultimately higher grain and biological yield. This depicts that nitrogen management based on Leaf Color Chart was responded to positively by maize hybrids. Moreover, the assessment of agrometeorological indices revealed the facts related to the impact of temperature and solar radiation on the phenology of crop, yield and heat energy consumption in maize hybrids. The regression analysis further stated the impact of different agrometeorological indices on the grain and biological yields of maize. Thus, it can be concluded that Nitrogen management based on LCC $\leq$ 5@30 kg N ha$^{-1}$ should be adopted in maize hybrids especially in Shalimar Maize Hybrid-2 to attain higher yield in the temperate climate of Kashmir valley.

**Supplementary Materials:** The following supporting information can be downloaded at: https://www.mdpi.com/article/10.3390/agronomy12122981/s1. Figure S1: Different LCC shades used for nitrogen application in maize varieties.

**Author Contributions:** Conceptualization, S.F. and T.A.B.; methodology, S.F., R.H.K., T.A.B. and A.N., software, M.S.M., A.N. and T.A.B.; validation, T.A.B., R.H.K., M.S.M., A.N. and S.F.; formal analysis, I.A.-A., A.E.S., M.V., A.M., R.I., T.A.B. and M.S.M.; investigation, M.S.M., S.F. and R.H.K.; resources, I.A.-A., A.E.S., A.M., R.I., M.V., S.A.A. and A.N.; writing—original draft preparation, S.F., T.A.B., S.A.A. and M.S.M.; writing—review and editing, M.V., A.E.S., I.A.-A., M.V., R.I., A.M., S.A.A., R.H.K., M.S.M., S.F. and A.N. All authors have read and agreed to the published version of the manuscript.

**Funding:** The paper was funded by the Researchers Supporting Project number (RSP-2021/298), King Saud University, Riyadh, Saudi Arabia.

**Institutional Review Board Statement:** Not applicable.

**Informed Consent Statement:** Not applicable.

**Data Availability Statement:** Not applicable.

**Acknowledgments:** The authors extend their appreciation to the Researchers Supporting Project number (RSP-2021/298), King Saud University, Riyadh, Saudi Arabia. The authors express the deepest appreciation to the Division of Agronomy; Faculty of Agriculture, Sher-e-Kashmir University of Agricultural Sciences and Technology of Kashmir, Wadura, Sopore-193201, India for providing all the necessary facilities, suggestions, help, cooperation and praise to complete the research.

**Conflicts of Interest:** The authors declare no conflict of interest.

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
