# Peer review of "Leaf Color Chart (LCC)-Based Precision Nitrogen Management for Assessing Phenology, Agrometeorological Indices and Sustainable Yield of Hybrid Maize Genotypes under Temperate Climate"

_agronomy, doi:10.3390/agronomy12122981_

Round 1

Reviewer 1 Report

The authors propose a manuscript titled “Leaf color chart based precision nitrogen management for assessing phenology, agrometeorological indices and yield of hybrid maize genotypes under temperate ecology”. 

I suggest the following changes:

Abstract

My suggestion to the authors is adding a contextual sentence about their motivation for the study at the Abstract section

Materials and Methods

Try to describe the hybrids used more detailed

Conclusions

Regarding the parameters studied more detailed conclusion could be drawn

Author Response

Dear reviewer, we are highly thankful for your valuable suggestion and all suggestions/corrections were incorporated into the revised manuscript

Reviewer 2 Report

The manuscript entitled “Leaf color chart based precision nitrogen management for assessing phenology, agrometeorological indices and yield of hybrid maize genotypes under temperate ecology” used a two-year field experiment and three hybrid maize cultivars to investigate the effects of precision nitrogen management on growth and yield formation of maize. The work is interesting. My suggestion is minor revision.

1. Line 4: “temperate ecology” should be “temperate condition” or “temperate climate”

2. It's best to use the same words in the whole manuscript. “genotypes” or “cultivars”

3. Line 141: What is DAP and MOP?  Authors should mention the full name when firstly used abbreviation.

4. Can you provide the information of the machines used in LCC Based nitrogen application.

5. What is q ha-1? The unit of crop yield normally was expressed as t ha-1.

6. Can you provide a picture or photo of LCC in the manuscript?

Author Response

(The authors gave the same response as above.)

Reviewer 3 Report

Dear authors

I read the Leaf color chart based precision nitrogen management for assessing phenology, agrometeorological indices and yield of hybrid maize genotypes under temperate ecology article. It is an interesting subject.

General comments

Please answer my questions:

What is the novelty of this study?

Please edit the title. based on...

Special comments:

The abstract needs English editing

Line 55-56 need to revision

Line 76-79 need to revision

Line 146-147 need to revision

Line 152-154 need to revision not clear

Line 281-283 are not clear

Line 439-442 are not clear

Author Response

(The authors gave the same response as above.)

Reviewer 4 Report

Tools  like Leaf Colour Chart (LCC) ... Define abbreviations at first mention.

 in attaining different phenophases during both the years. 

...assess the impact of precision nitrogen management on the phenology, yield, and agrometeorological indices of hybrid maize cultivars at the Agronomy Research Farm, ... (add comma)

In the case of control...  replace by:  In the control case

The crop attained maturity in greater number of days when nitrogen was applied through the LCC score-based: 

It would be interesting to describe other variations in the phenotype when the treatment is applied. e.g. Number of grains per cob or 1000-grains weight.

 Grain and biological yield: Delve into how they were determined.

Author Response

(The authors gave the same response as above.)

Reviewer 5 Report

How about changing the title? "Color chart of corn leaves based on precision nitrogen management for evaluation of phenology, agrometeorological indices and yield"

Line 78 - Based on which bibliographic reference?

Line 144 - why was large line spacing used? Commercial plantations in Brazil are used with spacings of 50 cm x 25 cm. Population of 80000 plants per hectare.

Line 147 - What type of irrigation management was used? Via climate, via soil? Has the humidity been raised to field capacity? Which irrigation system? Center pivot, sprinkler, drip?

RESULTS: I believe there was a lack of data on the irrigation depth applied in the two years of cultivation. With this information, added to the precipitation, it is possible to estimate the efficiency of water use in the two crops. Important and simple information to obtain. It can further substantiate the discussion of the work.

Line 386 - To further support the discussion of the work, water use efficiency fits very well here. It enriches the discussion.

REFERENCES: Of a total of 48 citations, 34 are old, more than 5 years old. I believe that an improvement in references is needed, replacing old citations with newer citations.

Author Response

(The authors gave the same response as above.)

Round 2

Reviewer 3 Report

Dear authors 

I accept your revision

best regards